# Prevalence of dental caries in the primary, mixed and permanent dentitions in Nigeria: A systematic review and meta-analysis

Folahanmi Tomiwa Akinsolu[1,2,3,4☺]*, Olunike Rebecca Abodunrin[1,2,5☺], Abel Chukwuemeka[1,2], Mobolaji Timothy Olagunju[5], Ifeoluwa Eunice Adewole[1,2‡], Abideen Olurotimi Salako[1,2,3‡], George Uchenna Eleje[1,6,7‡], Adebola Oluyemisi Ehizele[1,8☺], Joanne Lusher[1,9‡], Foluso Owotade[1,10‡], Oliver Chukwujekwu Ezechi[1,2,3☺], Morẹ́nikẹ́ Oluwátóyìn Foláyan[1,10,11☺]

1 Oral Health Initiative, Center for Reproduction and Population Health Studies, Nigerian Institute of Medical Research, Yaba, Lagos, Nigeria, 2 Department of Public Health, Faculty of Basic Medical and Health Sciences, Lead City University, Ibadan, Oyo, Nigeria, 3 Clinical Sciences Department, Nigerian Institute of Medical Research, Yaba, Lagos, Nigeria, 4 Global Health and Infectious Diseases, University of North Carolina at Chapel Hill, Chapel Hill, North Carolina, United States of America, 5 Nanjing Medical University, Nanjing, China, 6 Department of Obstetrics and Gynaecology, Nnamdi Azikiwe University Teaching Hospital Nnewi, Nnewi, Nigeria, 7 Effective Care Research Unit, Department of Obstetrics and Gynaecology, Nnamdi Azikiwe University, Awka, Nigeria, 8 Department of Periodontics, School of Dentistry, College of Medical Sciences, University of Benin, Benin City, Nigeria, 9 Provost's Group, Regent's University London, London, United Kingdom, 10 Department of Oral Medicine and Oral Pathology, Obafemi Awolowo University, Ile-Ife, Nigeria, 11 Department of Child Dental Health, Obafemi Awolowo University, Ile-Ife, Nigeria

☺ These authors contributed equally to this work.
‡ IEA, AOS, GUE, JL, and FO also contributed equally to this work.
* Folahanmi.tomiwa@gmail.com

## Abstract

### Background

The national prevalence of dental caries in Nigeria is currently unknown. The objective of this systematic review and meta-analysis was to determine the prevalence of dental caries in the primary, mixed and permanent dentition among residents in urban, rural and semi-urban Nigeria.

### Methods

A systematic search was conducted in PubMed, Web of Science, Scopus, CINAHL and Embase databases and Google Scholar for studies published between January 2001 and December 2023 reporting the prevalence of dental caries in Nigeria. The review was registered with PROSPERO (CRD42022362019) and conducted in accordance with the PRISMA guidelines. Data extracted included study design, sample size, age of participants, and study location. Study quality and risk of bias were assessed. A random-effects meta-analysis was performed to estimate pooled prevalence.

**Data availability statement:** All relevant data are within the manuscript and its Supporting information files.

**Funding:** Grant Number: 5NM-ADJGT-22-0082 from the Nigerian Institute for Medical Research. The funders had no role in study design, data collection and analysis, decision to publish, or preparation of the manuscript.

**Competing interests:** The authors have declared that no competing interests exist.

## Results

A total of 1,010 records were identified, of which 52 studies were included in the systematic review and 35 were eligible for meta-analysis. Most studies were conducted in Southwestern Nigeria. The overall pooled prevalence of dental caries in Nigeria was 17% (95% CI: 14%–21%; $I^2 = 97\%$). The pooled prevalence was 16% (95% CI: 10%–24%; $I^2 = 98\%$) in primary dentition, 16% (95% CI: 11%–23%; $I^2 = 97\%$) in mixed dentition, and 20% (95% CI: 16%–26%; $I^2 = 96\%$) in permanent dentition. By setting, the pooled prevalence was 22% (95% CI: 7%–52%; $I^2 = 98\%$) in rural areas, 17% (95% CI: 14%–22%; $I^2 = 97\%$) in semi-urban areas, and 14% (95% CI: 6%–29%; $I^2 = 98\%$) in urban areas. Substantial heterogeneity was observed across studies.

## Conclusion

Dental caries remains a significant public health concern in Nigeria, affecting approximately one in five individuals. Although variations were observed across dentition types and geographic settings, substantial heterogeneity indicates diverse epidemiological contexts across the country. Strengthened preventive strategies and improved access to oral healthcare services are needed to address the burden of dental caries nationwide.

## Introduction

Dental caries is a worldwide health issue [1–4], affecting approximately 2.3 billion people with caries in permanent teeth and 514 million children with caries in primary teeth worldwide [2]. Dental caries represent a lifelong health burden, beginning in childhood and potentially persisting into adulthood if untreated [3]. Among children, dental caries is one of the most common conditions affecting the deciduous dentition [5,6], and globally, approximately 34% of the population has untreated cavities in permanent teeth [2]. These figures underscore the need for coordinated global responses tailored to context-specific risk factors and health system capacities.

The burden of dental caries is particularly concerning in low- and middle-income countries (LMICs), where prevalence remains high and continues to increase [4,7]. Many LMICs face significant challenges, including limited access to dental care, inadequate resources, and disparities in oral health education [8,9]. Untreated dental caries has far-reaching consequences for health and well-being. It is associated with pain, difficulty chewing, and dietary restrictions, which may contribute to nutritional deficiencies [10–16]. Beyond physical health, dental caries also has psychosocial implications, including social stigma, reduced self-esteem, social withdrawal, and poorer mental health outcomes [17–19]. The economic burden is substantial as well, encompassing treatment costs, missed workdays, and reduced productivity [20]. These challenges are compounded in settings with limited oral health insurance coverage [4,21], low awareness of preventive oral healthcare [22], and a high prevalence of poor mental health and reduced quality of life [23].

Nigeria, as an LMIC, faces similar challenges. Reports indicate a considerable burden of untreated dental caries in both permanent and deciduous dentition [23]. Despite increasing recognition of oral health as an important component of overall health, comprehensive national data on the prevalence of dental caries in Nigeria remain limited [24]. Data from the Global Burden of Disease Study show that between 1990 and 2017, the number of persons with untreated caries in Nigeria increased by 22.5% in primary dentition and by 91.5% in permanent dentition [25]. Although Nigeria ranked as the 20th African country with the lowest percentage change in oral disease prevalence during this period [25], these increases highlight the need for updated and nationally synthesized epidemiological evidence. Establishing reliable baseline prevalence estimates is essential for informing strategic planning, prevention programs, and policy decisions related to caries control [26].

Given these gaps, a comprehensive synthesis of existing evidence is necessary to better understand the epidemiology of dental caries in Nigeria. By integrating data across different regions and examining variations by dentition type and geographical setting (urban, rural, and semi-urban), this study provides a clearer picture of the burden and distribution of dental caries in the country. Such evidence is critical for guiding targeted public health interventions, improving resource allocation, and strengthening oral health policies.

This systematic review and meta-analysis therefore aim to determine the pooled prevalence of dental caries in primary, mixed, and permanent dentition among the Nigerian population and to explore differences in prevalence by geographical location.

## Methods

### Study protocol

This systematic review and meta-analysis was registered with PROSPERO (CRD42022362019). The study was reported following the Preferred Reporting Item for Systematic Reviews and Meta-analyses (PRISMA) statement and checklist [27] (See S1 File).

### Research question

What is the prevalence and severity of dental caries in the primary, mixed, and permanent dentition among residents in Nigeria, and how does this vary by geographical location (urban, rural, semi-urban)?

The PICO framework was used to understand the prevalence and severity of dental caries in Nigeria [27]. Table 1 shows the components of the PICO framework used for this study.

### Search strategy and selection of studies

PubMed, Web of Science, Scopus, CINAHL, and Google Scholar were searched from January 2001 to December 2023. The search targeted studies focused on dental caries and tooth decay in various types of teeth, in the Nigerian population, with emphasis on epidemiological concepts such as prevalence, burden, incidence, severity, and demographics. The search terms used for the search from the databases with the Boolean operators "OR" and "AND" are:

**Table 1. PICO component.**

| Component | Description |
| --- | --- |
| Population (P) | Residents of Nigeria (children and adults) in urban, rural, and semi-urban areas |
| Intervention (I) | Not applicable (Observational study) |
| Comparison (C) | Comparison between different dentition types (primary, mixed, permanent) and geographic settings (urban, rural, semi-urban) |
| Outcome (O) | Prevalence and severity of dental caries |

("dental caries," OR "tooth decay," OR "caries," OR "tooth caries," OR "tooth decay") AND ("primary teeth," OR "deciduous teeth," OR "milk teeth," OR "baby teeth," OR "permanent teeth," OR "mixed dentition") AND ("Nigeria," OR "Nigerian") AND ("prevalence," OR "burden of disease," OR "severity of illness index," OR "incidence," OR "epidemiology," OR "demography"). The detailed search strategy are described in S2 File.

The search outcomes in the database were downloaded to the reference management software EndNote X9 and duplicate items were sorted out and removed. Two authors (O.R.A. and M.T.O.) independently assessed the title and abstracts of each study to assess if it met the inclusion criteria. Studies that did not meet the inclusion criteria were excluded. Two authors (O.M.O. and I.E.A.) independently assessed the eligibility of the retrieved papers and resolved disagreements by discussion or recourse to a third author (A.C. or F.T.A.).

## Eligibility criteria

All published and unpublished studies, including grey literature reporting on the prevalence of dental caries in Nigeria's permanent and deciduous dentition, were eligible for study inclusion. These included cross-sectional, cohort, and case-control studies. For non-cross-sectional studies, inclusion in the meta-analysis was limited to studies that reported extractable prevalence-related data at a defined assessment point relevant to the review objective. There was no language restriction.

Studies were excluded if they did not provide information on the sample size, had inaccurate or unavailable outcome data, had no data on the prevalence of dental caries, and featured duplicate samples. Furthermore, review articles, studies with overlapping data from other included studies, case reports, case series, editorials, laboratory investigations, or reviews devoid of primary data were also excluded. The detailed excluded studies and the reasons are reported in S3 File.

## Data extraction

Four independent authors (I.E.A., M.T.O., A.C., and O.R.A.) used a pretested data extraction form prepared in Microsoft Excel to independently extract details of articles that met the inclusion criteria. The information was the author's name, year of article publication, study design (cross-sectional, cohort-based), study location, and study setting were recorded. Information on the study participants (sample size and age range) was extracted. The prevalence of dental caries in each study was also extracted. Discrepancies among reviewers during the extraction process were resolved by a fifth author (F.T.A.).

## Quality and risk of bias assessment

The quality of all the included articles was assessed by two independents researchers (O.R.A. and M.T.O.). Joanna Briggs Institute (JBI) critical appraisal checklist for prevalence studies was used to establish risk of biasness in included article [28]. JBI appraisal checklist for prevalence studies is based on 9 questions: (1) Was the sample frame appropriate to address the target population? (2) Were the study participants sampled in an appropriate way? (3) Was the sample size adequate? (4) Were the study subjects and the setting described in detail? (5) Was the data analysis conducted with sufficient coverage of the identified sample? (6) Were valid methods used for the identification of the condition? (7) Was the condition measured in a standard, reliable way for all participants? (8) Was there appropriate statistical analysis? (9) Was the response rate adequate, and if not, was the low response rate managed appropriately? [28]. Each of the question was analyzed by giving score 1 or 0 (yes = 1), (no = 0), and (unclear or not applicable = 0). The over-all score for each included study was presented as percentages and study was characterized according to different degrees of risk of bias (high risk of bias if 20–50% items scored yes, moderate risk of bias if 50–80% items scored yes, and low risk of bias if 80–100% items scored yes as per JBI checklist). (See S1 Table)

## GRADE rating quality of evidence

The GRADE (Grading of Recommendations, Assessment, Development, and Evaluation) approach is a structured and transparent method for assessing the quality of evidence and the strength of recommendations in healthcare research. This approach evaluates the quality of evidence based on several criteria, including study design, consistency of findings, directness of evidence, precision of estimates, and risk of bias. The evidence is then rated into four categories: high, moderate, low, or very low.

In this study, the GRADE approach was applied to assess the quality of evidence from the included studies critically. This involved a systematic review of each study, applying the GRADE criteria rigorously to ensure that the conclusions drawn are based on reliable evidence. GRADE ensures that any recommendations are supported by the highest quality evidence available and are relevant to real-world clinical settings. Further details about the GRADE methodology can be found in the GRADEpro software, which is freely accessible via Cochrane resources (http://tech.cochrane.org/revman/gradepro). (See S2 Table)

## Dealing with missing data

In this systematic review, no missing data were identified across the included studies. As a result, no additional efforts were required to contact primary authors or impute missing data. All data reported in the studies were complete and were analyzed as presented.

## Data synthesis process

The data included in the studies were synthesized using qualitative and quantitative approaches. In the qualitative synthesis, key characteristics of the included studies, such as study design, population demographics, and geographic location, were summarized to provide an overview of dental caries prevalence in Nigeria. The studies were grouped by their settings (urban, rural, or semi-urban) and dentition types (primary, mixed, or permanent), and patterns were identified across different regions of the country. Participant characteristics, such as age groups and sample sizes, were also summarized. The synthesis helped identify gaps in the literature, including underrepresentation from certain regions, and provided a broader understanding of the factors influencing dental caries prevalence in Nigeria.

## Statistical analysis

All statistical analyses were conducted using R software (RStudio environment). The pooled prevalence of dental caries was estimated using a random-effects meta-analysis based on the DerSimonian and Laird method. A random-effects approach was selected a priori due to anticipated clinical and methodological heterogeneity across studies, including differences in study populations, geographic regions, and diagnostic criteria [29,30].

For studies that did not report standard errors (S.E.), these were calculated from the reported prevalence and sample size using standard formulas. Prevalence estimates were logit-transformed where appropriate to stabilize variances prior to pooling. Pooled prevalence estimates with 95% confidence intervals (CIs) were then back-transformed for presentation.

Statistical heterogeneity was assessed using Cochran's Q test ($p < 0.10$ indicating significant heterogeneity) and quantified using the $I^2$ statistic, with values of approximately 25%, 50%, and ≥75% interpreted as low, moderate, and high heterogeneity, respectively. The between-study variance ($\tau^2$) was also reported [29,30].

Subgroup analyses were conducted according to geographical setting and dentition type. Differences between subgroups were evaluated using the chi-square test for subgroup differences.

Meta-regression analysis was performed using publication year as a continuous moderator to evaluate potential temporal trends in prevalence estimates.

Publication bias and small-study effects were assessed through visual inspection of funnel plots and formally evaluated using Egger's linear regression test. A $p$-value $< 0.05$ was considered indicative of statistically significant small-study effects.

Sensitivity analyses were performed to assess the robustness of the pooled prevalence estimate. First, a leave-one-out influence analysis was conducted by sequentially excluding each study and recalculating the pooled prevalence to determine whether any individual study disproportionately influenced the overall estimate. Second, a trim-and-fill analysis was performed in R to evaluate the potential impact of missing studies due to publication bias and to generate an adjusted pooled prevalence estimate. In addition, sensitivity analysis was used to evaluate the influence of study design on the pooled estimate, including the effect of excluding the single retrospective/cohort-type study included in the meta-analysis.

All statistical tests were two-sided, and statistical significance was set at $p < 0.05$ unless otherwise specified.

## Ethical approval

Ethical approval was not required for this systematic review as the research was based on information retrieved from published studies.

## Results

### Selection of studies

As shown in Fig 1, 1,010 records were retrieved. After removing duplicates, 995 records remained for eligibility screening based on title and abstract. Of these, 886 studies were excluded based on title and abstract. After reviewing the full-text records of 107 studies (106 from the search on databases and a study from a citation search), 52 met the inclusion criteria [20,31–97].

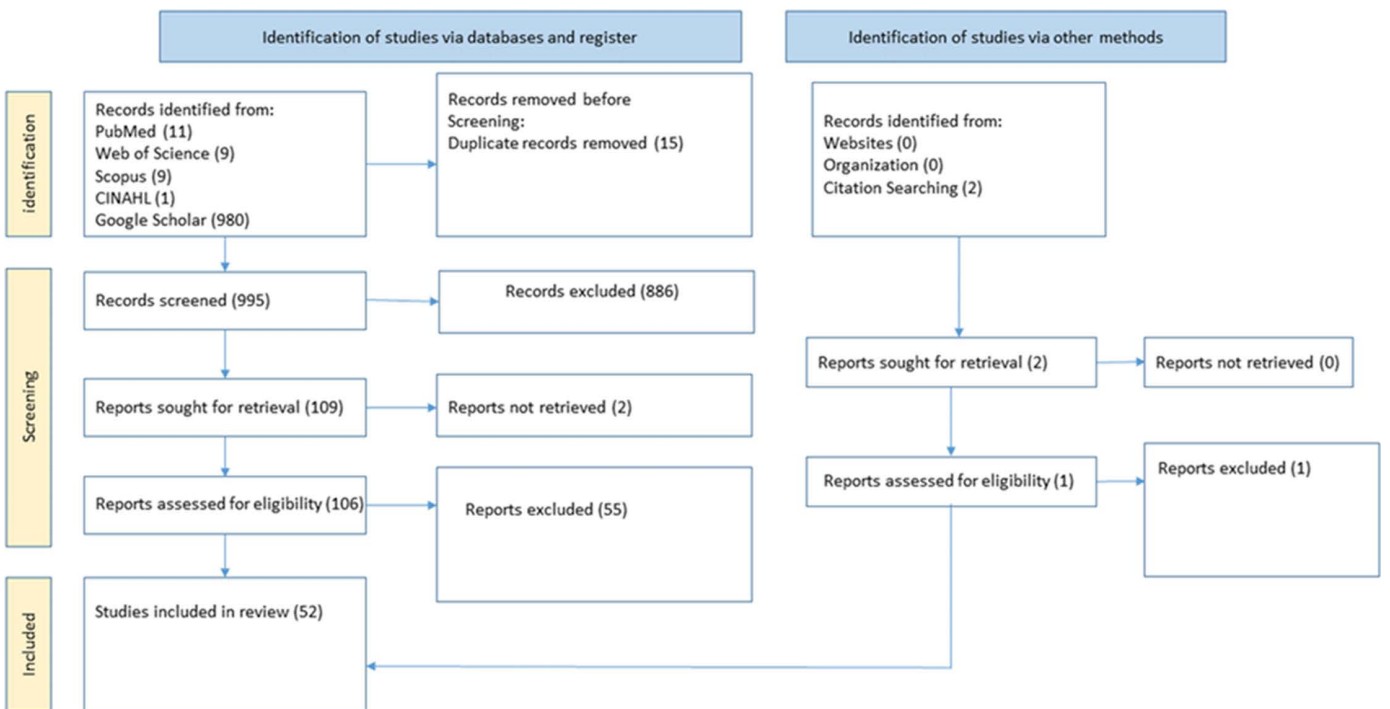

**Fig 1. PRISMA Flow Chart to identify studies on the prevalence of caries in Nigeria.**

## Characteristics of included studies

The extracted data from the 52 studies are presented in Table 2. The data on dental caries prevalence was generated between 2006 and 2022. There were four (7.7%) studies published between 2005 and 2010 [32,33,45,79], 35 (67.3%) studies published between 2011 and 2020 [31,34,35,37–44,46–48,51–59,64,66,68,69,74–78,80,81], and 13 (25.0%) studies published between 2021 and 2022 [36,49,50,60–63,65,67,70–73].

The sample size of the included studies ranged from 17 to 2,835 with an aggregated number of 38,253 study participants. There were 48 (92.3%) cross-sectional studies [13,20,25,26,31–53,55–76,78–83,85–92,94,95] and 4 cohort studies [36,43,54,77].

Most of the studies were conducted in Southwest Nigeria: 14 studies in Lagos State [31,32,34,35,48,54,65–68,73,77–80,92], 11 studies in Osun State [33,37,49–53,56,57,75–77,85–89,94,96], seven studies in Oyo State [20,43–46,55,58,72], and a study in Ogun State [38] respectively. Some other studies were conducted in South-south Nigeria: four in Edo State [36,40,42,59,60,82,83] and three in Rivers State [41,47,64,84]. Four studies were conducted in Enugu, in the southeastern part of Nigeria [62,69–71,91]. The remaining studies were conducted in the Northern Territory part of Nigeria: a study in Kwara State [81,90,95] in Northcentral Nigeria, and three studies in Northwestern Nigeria: two studies in Kano State [39,63], and a study in Sokoto State [61]. There is no data from Northeastern Nigeria.

In addition, 21 (40.4%) articles demonstrated a low risk of bias [20,32,34,38,40,42–45,49–52,57,59,62,63,72,75,76,81,83–89,94,96], 30 (57.7%) articles had moderate-risk of bias category [33,35–37,39,41,46–48,53–56,58,60,61,64–71,73,74,77–80] and a study had high risk of bias [31,95].

Also, 49 (94.2%) of the studies used the Decayed, Missing, and Filled Teeth index for permanent teeth (dmft/DMFT) index to assess dental caries [20,31–35,37–61,64,65,67–81]. A study used the International Caries Detection and Assessment System (ICDAS) index to assess dental caries [36].

## Quality assessment

The studies were generally categorized into three quality levels: low, moderate, and high risk of bias. Most studies 40.4% (21/52) had low risk of bias [20,32,34,38,40,42–45,49–52,57,59,62,63,72,75,76,81] while 57.7% (30/52) had a moderate risk of bias [35–39,41,46–48,53–56,58,60,61,64–71,73,74,77–80]. Studies with low risk of bias generally met key criteria like adequate sample size, detailed descriptions of the population and setting, appropriate sampling techniques, and valid and reliable methods for measuring the condition of interest. On the other hand, a study had a high risk of bias [31] due to weaknesses in sample size adequacy and the use of less reliable condition identification methods. (See S3 File)

Table 3 shows the profile of the 35 studies included in the meta-analysis. These were 13 studies showing the prevalence of dental caries in primary dentition, with dental caries prevalence ranging from 1.37% [71] to 95.6% [65]. Twelve studies showed dental caries' prevalence in mixed dentition, with dental caries prevalence ranging from 5.20% [50] to 29.2% [67]. Nineteen studies showed dental caries' prevalence in permanent dentition, ranging from 1.16% [38] to 64.7% [43]. The sample size for the studies in the primary dentition was 7,932, for the mixed dentition was 9,223, and for the permanent dentition was 10,729.

In addition, 20 (38.5%) studies included were conducted in the urban setting [32,35,40–46,53,55,58,62,65–71], three (5.77%) were conducted in rural settings [38,61,64] and 12 (23.1%) were conducted in semi-urban settings [33,37,48,50–52,56,57,63,72,78,79]. The prevalence of dental caries ranged from 6.60% [53] to 82.1% [65] in the urban setting, 12.3% [38] to 35.6% [61] in the rural setting, and 3.3% [52] to 24.7% [72].in the semi-urban setting.

Furthermore, of the 13 studies [32,46,51–53,55,57,65–67,69,70,79] in the primary dentition, 9 were conducted in the urban setting with prevalence ranging from 1.37% to 95.60% [51,52,57,79], 4 were conducted in the semi-urban setting

**Table 2. Characteristics of included studies.**

| S/No | First Author (Year) | Title | Study type | Sample Size | Age range | Dental Caries Index | State | Quality Score |
|------|---------------------|-------|-----------|-------------|-----------|---------------------|-------|---------------|
| 1. | Adekoya-Sofowora, et al., 2006 [33] | Dental caries in 12-year-old suburban Nigerian school children | Cross-Sectional | 402 | 12 years | DMFT | Osun | Moderate |
| 2. | Adeniyi, et al., 2009 [32] | Dental caries occurrence and associated oral hygiene practices among rural and urban Nigerian pre-school children | Cross-Sectional | 404 | 18 - 60 months | dmft | Lagos | Low |
| 3. | Adeniyi, et al., 2012 [34] | Prevalence and Pattern of Dental Caries Among a Sample of Nigerian Public Primary School Children | Cross-Sectional | 2,835 | 5 - 16 years | DMFT | Lagos | Low |
| 4. | Adeniyi, et al., 2016 [31] | Dental Caries and Nutritional Status of School Children in Lagos, Nigeria – A Preliminary Survey | Cross-Sectional | 973 | 5 - 10 years | DMFT | Lagos | High |
| 5. | Adeniyi, et al., 2017 [35] | Self-Reported Dental Pain and Dental Caries Among 8–12-Year-Old School Children: An Exploratory Survey in Lagos, Nigeria | Cross-Sectional | 414 | 8 - 12 years | DMFT | Lagos | Moderate |
| 6. | Ajayi, et al., 2015 [43] | A 5-year retrospective study of rampant dental caries among adult patients in a Nigerian Teaching Hospital | Cohort | 17 | 22 - 61 years | DMFT | Oyo | Low |
| 7. | Akhigbe, et al., 2022 [36] | Age-specific associations with dental caries in HIV-infected, exposed but uninfected and HIV-unexposed uninfected children in Nigeria | Cohort | 544 | 4 - 11 years | ICDAS Criteria | Edo | Moderate |
| 8. | Akinwonmi, et al., 2019 [37] | Oral health characteristics of children and teenagers with special health care needs in Ile-Ife, Nigeria. | Cross-Sectional | 206 | 6 - 19 years | dmft/DMFT | Osun | Moderate |
| 9. | Akinyamoju, et al., 2018 [38] | Dental Caries and Oral Hygiene Status: Survey of Schoolchildren in Rural Communities, Southwest Nigeria | Cross-Sectional | 778 | 7 - 17 years | DMFT | Ogun | Low |
| 10. | Aliyu, et al., 2019 [39] | Prevalence of dental caries in children with chronic heart disease | Cross-Sectional | 260 | 1 - 14 years | dmft/DMFT | Kano | Moderate |
| 11. | Braimoh, et al., 2011 [40] | Caries and periodontal health status of prison inmates in Benin City, Nigeria. | Cross-Sectional | 140 | 18 - 64 years | DMFT | Edo | Low |
| 12. | Braimoh, et al., 2014 [41] | Caries Distribution, Prevalence, and Treatment Needs among 12–15-Year-Old Secondary School Students in Port Harcourt, Rivers State, Nigeria | Cross-Sectional | 195 | 12 - 15 years | DMFT | Rivers | Moderate |
| 13. | Chukwumah, et al., 2015 [42] | Impact of dental caries and its treatment on the quality of life of 12- to 15-year-old adolescents in Benin, Nigeria | Cross-Sectional | 1,790 | 12 - 15 years | DMFT | Edo | Low |
| 14. | Dedeke, et al., 2014 [44] | Findings from a study in a defined urban population in South- western Nigeria using the PUFA index | Cross-Sectional | 2,149 | 6 – years | dmft/DMFT | Oyo | Low |
| 15. | Denloye, et al., 2005 [45] | A Study of dental caries prevalence in 12–14-year-old school children in Ibadan, Nigeria | Cross-Sectional | 577 | 12 - 14 years | DMFT | Oyo | Low |
| 16. | Denloye, et al., 2012 [46] | Oral health status of children seen at a pediatric neurology clinic in a tertiary hospital in Nigeria | Cross-Sectional | 61 | > 1 Year | dmft | Oyo | Moderate |
| 17. | Eigbobo, et al., 2017 [47] | Dental caries experience in primary school pupils in Port Harcourt, Nigeria | Cross-Sectional | 430 | 3 - 12 years | dmft/DMFT | Rivers | Moderate |
| 18. | Ekowmenhenhen, et al., 2019 [48] | Adult Dental Caries Experience: A Rural-Urban Comparison in South-western Nigeria | Cross-Sectional | 474 | 18 - 64 years | DMFT | Lagos | Moderate |
| 19. | El Tantawi, et al., 2021 [49] | Association between mental health, caries experience and gingival health of adolescents in sub-urban Nigeria | Cross-Sectional | 1,234 | 10 - 19 years | DMFT | Osun | Low |

*(Continued)*

**Table 2.** (Continued)

| S/No | First Author (Year) | Title | Study type | Sample Size | Age range | Dental Caries Index | State | Quality Score |
|------|---------------------|-------|------------|-------------|-----------|---------------------|-------|---------------|
| 20. | Folayan, et al., 2012 [54] | Caries incidence in a cohort of primary school students in Lagos State, Nigeria followed up over a 3 years period | Cohort | 192 | 2 - 10 years | dmft/DMFT | Lagos | Moderate |
| 21. | Folayan, et al., 2015 [53] | Prevalence, and early childhood caries risk indicators in preschool children in suburban Nigeria | Cross-Sectional | 497 | 6 - 71 months | dmft | Osun | Moderate |
| 22. | Folayan, et al., 2020 [52] | Malnutrition, enamel defects, and early childhood caries in preschool children in a sub-urban Nigeria population | Cross-Sectional | 1,549 | 6 - 71 months | dmft | Osun | Low |
| 23. | Folayan, et al., 2020 [51] | Validation of maternal report of early childhood caries status in Ile-Ife, Nigeria | Cross-Sectional | 1155 | 0 - 6 years | dmft | Osun | Low |
| 24. | Folayan, et al., 2022 [50] | Risk indicators for dental caries, and gingivitis among 6–11-year-old children in Nigeria: a household-based survey | Cross-Sectional | 1326 | 6 - 11 years | dmft/DMFT | Osun | Low |
| 25. | Iyun, et al., 2014 [55] | Prevalence and pattern of early childhood caries in Ibadan, Nigeria | Cross-Sectional | 540 | 3-5 years | dmft | Oyo | Moderate |
| 26. | Kolawole, et al., 2016 [57] | Digit Sucking Habit and Association with Dental Caries and Oral Hygiene Status of Children Aged 6 Months to 12 Years Resident in Semi-Urban Nigeria | Cross-Sectional | 992 | 6 months – 12 years | dmft/DMFT | Osun | Low |
| 27. | Kolawole, et al., 2019 [56] | Association between malocclusion, caries and oral hygiene in children 6–12 years old resident in suburban Nigeria | Cross-Sectional | 495 | 6 - 12 years | dmft/DMFT | Osun | Moderate |
| 28. | Lawal, et al., 2017 [58] | Dental caries experience and treatment needs of an adult female population in Nigeria. | Cross-Sectional | 430 | 16 - 59 years | DMFT | Oyo | Moderate |
| 29. | Lawal, et al., 2019 [20] | Impact of Untreated Dental Caries on Daily Performances of Children from Low Social Class in an Urban African Population: The Importance of Pain | Cross-Sectional | 1,286 | 6 - 15 years | dmft/DMFT | Oyo | Low |
| 30. | Nnawuihe, et al., 2016 [59] | An assessment of dental caries and periodontal disease burden in selected primary and secondary school children in Edo State, Southern – Nigeria | Cross-Sectional | 2066 | 4 - 21 years | dmft/DMFT | Edo | Low |
| 31. | Nnawuihe, et al., 2021 [60] | Oral Disease Burden amongst Residents of an Internally Displaced Persons Camp in Nigeria | Cross-Sectional | 437 | 4 - 71 years | DMFT | Edo | Moderate |
| 32. | Ogbeide, et al., 2022 [61] | Prevalence of Dental Caries Among Children and Young Adults with Disabilities Attending a Special Needs School in Sokoto, Nigeria | Cross-Sectional | 236 | 6 - 28 years | dmft/DMFT | Sokoto | Moderate |
| 33. | Okoli, et al., 2021 [62] | Prevalence of common oral diseases among Senior Secondary School students in Enugu State, Nigeria | Cross-Sectional | 900 | 12 - 23 years | – | Enugu | Low |
| 34. | Okolo, et al., 2022 [63] | Dental Caries Prevalence, Severity, and Pattern Among Male Adolescents in Kano, Nigeria | Cross-Sectional | 694 | 10 - 12 years | | Kano | Low |
| 35. | Olabisi, et al., 2015 [64] | Prevalence of dental caries and oral hygiene status of a screened population in Port Harcourt, Rivers State, Nigeria | Cross-Sectional | 288 | 20 - 64 years | DMFT | Rivers | Moderate |
| 36. | Olatosi, et al., 2015 [66] | The prevalence of early childhood caries and its associated risk factors among preschool children referred to a tertiary care institution | Cross-Sectional | 302 | 6 - 71 months | – | Lagos | Moderate |

*(Continued)*

**Table 2.** (Continued)

| S/No | First Author (Year) | Title | Study type | Sample Size | Age range | Dental Caries Index | State | Quality Score |
|---|---|---|---|---|---|---|---|---|
| 37. | Olatosi, et al., 2020 [68] | Disparities in Caries Experience and Socio-Behavioral Risk Indicators Among Private School Children in Lagos, Nigeria | Cross-Sectional | 592 | 5 - 16 years | dmft/DMFT | Lagos | Moderate |
| 38. | Olatosi, et al., 2022 [65] | Dental Caries Severity and Nutritional Status of Nigerian Preschool Children | Cross-Sectional | 273 | 1 - 6 years | dmft/DMFT | Lagos | Moderate |
| 39. | Olatosi, et al., 2022 [67] | Dental caries and oral health: an ignored health barrier to learning in Nigerian slums: a cross sectional survey | Cross-Sectional | 684 | 6 - 11 years | DMFT | Lagos | Moderate |
| 40. | Onyejaka, et al., 2016 [69] | Risk Factors of Early Childhood Caries among Children in Enugu, Nigeria | Cross-Sectional | 429 | 0 - 5 Years | dmft | Enugu | Moderate |
| 41. | Onyejaka, et al., 2021 [71] | Prevalence and Associated Factors of Dental Caries among Primary School Children in South-East Nigeria | Cross-Sectional | 657 | 5 - 17 years | dmft/DMFT | Enugu | Moderate |
| 42. | Onyejaka, et al., 2021 [70] | Relationship Between Socio-Demographic Profile, Parity and Dental Caries Among a Group of Nursing Mothers in Southeast, Nigeria | Cross-Sectional | 408 | 15 - 52 years | DMFT | Enugu | Moderate |
| 43. | Osuh, et al., 2022 [72] | Prevalence and determinants of oral health conditions and treatment needs among slum and non-slum urban residents: Evidence from Nigeria | Cross-Sectional | 1357 | 18 years and above | DMFT | Oyo | Low |
| 44. | Oyedele, et al., 2018 [75] | Impact of oral hygiene and socio-demographic factors on dental caries in a suburban population in Nigeria | Cross-Sectional | 2,107 | 8 - 16 years | DMFT | Osun | Low |
| 45. | Oyedele, et al., 2020 [74] | Dental Caries Experience and MIH in Children Pattern and Severity NJBCS. | Cross-Sectional | 391 | 3 - 16 years | dmft/DMFT | South-south | Moderate |
| 46. | Oyeparo, et al., 2021 [73] | Association between dental caries, odontogenic infections, oral hygiene status and anthropometric measurements of children in Lagos, Nigeria | Cross-Sectional | 278 | 4 - 16 years | dmft/DMFT | Lagos | Moderate |
| 47. | Ozeigbe, et al., 2013 [76] | Prevalence and clinical consequences of untreated dental caries using PUFA index in suburban Nigerian school children | Cross-Sectional | 1,266 | 4 - 16 years | dmft/DMFT | Osun | Low |
| 48. | Sofola, et al., 2014 [77] | Changes in the prevalence of dental caries in primary school children in Lagos State, Nigeria | Cohort | 633 | 2 - 12 years | dmft/DMFT | Lagos | Moderate |
| 49. | Soroye, et al., 2016 [78] | Oral health status, knowledge of dental caries aetiology, and dental clinic attendance: A comparison of secondary school students in the rural and urban areas of Lagos | Cross-Sectional | 598 | 12 - 26 years | DMFT | Lagos | Moderate |
| 50. | Sowole, et al., 2007 [79] | Dental caries pattern and predisposing oral hygiene related factors in Nigerian preschool children | Cross-Sectional | 389 | 6 - 71 months | dmft | Lagos | Moderate |
| 51. | Umeizudike, et al., 2019 [80] | Oral health status and treatment needs of internally displaced persons | Cross-Sectional | 123 | Adults and Children | dmft/DMFT | Lagos | Moderate |
| 52. | Uthman, et al., 2018 [81] | Prevalence of dental caries in public and private primary schools in Ilorin South Local Government Area of Kwara State, Nigeria | Cross-Sectional | 800 | 5 - 15 years | DMFT | Kwara | Low |

ICDAS, International Caries Detection and Assessment System; dmft, Decayed, Missing and Filled Teeth index for primary teeth; DMFT, Decayed, Missing and Filled Teeth index for permanent teeth.

**Table 3. Studies included in the meta-analysis.**

| S/No | First Author (Year) | Types of Dentitions | Sample size | Prevalence of dental caries (%) | Study Settings |
|---|---|---|---|---|---|
| 1. | Adekoya-Sofowora, et al., 2006 [33] | Permanent | 402 | 13.9 | Semi-urban |
| 2. | Adeniyi, et al., 2009 [32] | Primary | 404 | 10.9 | Urban |
| 3. | Adeniyi, et al., 2017 [35] | Mixed | 414 | 21.0 | Urban |
| 4. | Ajayi, et al., 2015 [43] | Permanent | 17 | 64.71 | Urban |
| 5. | Akinwonmi, et al., 2019 [37] | Mixed Permanent | 206 | 15.0 7.80 | Semi-urban |
| 6. | Akinyamoju, et al., 2018 [38] | Mixed Permanent | 206 | 11.1 1.16 | Rural |
| 7. | Braimoh, et al., 2011 [40] | Permanent | 778 | 45.0 | Urban |
| 8. | Braimoh, et al., 2014 [41] | Permanent | 778 | 15.4 | Urban |
| 9. | Chukwumah, et al., 2015 [42] | Permanent | 140 | 21.9 | Urban |
| 10. | Dedeke, et al., 2014 [44] | Mixed | 195 | 9.31 | Urban |
| 11. | Denloye, et al., 2005 [45] | Permanent | 1,790 | 11.3 | Urban |
| 12. | Denloye, et al., 2012 [46] | Primary | 2,149 | 11.5 | Urban |
| 13. | Ekowmenhenhen, et al., 2019 [48] | Permanent | 577 | 17.5 | Semi-urban |
| 14. | Folayan, et al., 2015 [53] | Primary | 61 | 6.64 | Urban |
| 15. | Folayan, et al., 2020 [52] | Primary | 474 | 3.29 | Semi-urban |
| 16. | Folayan, et al.,2020 [51] | Primary | 497 | 7.97 | Semi-urban |
| 17. | Folayan, et al., 2022 [50] | Mixed | 1,549 | 5.20 | Semi-urban |
| 18. | Iyun, et al., 2014 [55] | Primary | 1155 | 23.5 | Urban |
| 19. | Kolawole, et al., 2016 [57] | Primary Mixed | 1326 | 3.02 7.46 | Semi-urban |
| 20. | Kolawole, et al., 2019 [56] | Mixed | 540 | 14.9 | Semi-urban |
| 21. | Lawal, et al., 2017 [58] | Permanent | 992 | 12.8 | Urban |
| 22. | Ogbeide, et al., 2022 [61] | Mixed Permanent | 992 | 13.1 22.5 | Rural |
| 23. | Okoli, et al., 2021 [62] | Permanent | 495 | 30.6 | Urban |
| 24. | Okolo, et al., 2022 [63] | Mixed | 430 | 22.9 | Semi-urban |
| 25. | Olabisi, et al., 2015 [64] | Permanent | 236 | 21.2 | Rural |
| 26. | Olatosi, et al., 2015 [66] | Primary | 236 | 16.1 | Urban |
| 27. | Olatosi, et al., 2020 [68] | Mixed Permanent | 900 | 11.3 4.73 | Urban |
| 28. | Olatosi, et al., 2022 [65] | Primary | 694 | 95.6 | Urban |
| 29. | Olatosi, et al., 2022 [67] | Primary Mixed Permanent | 288 | 36.1 29.2 16.8 | Urban |
| 30. | Onyejaka, et al., 2016 [69] | Primary | 302 | 9.79 | Urban |
| 31. | Onyejaka, et al., 2021 [71] | Primary Mixed Permanent | 592 | 1.37 19.8 1.52 | Urban |
| 32. | Onyejaka, et al., 2021 [70] | Permanent | 592 | 11.0 | Urban |
| 33. | Osuh, et al., 2022 [72] | Permanent | 684 | 24.7 | Semi-urban |
| 34. | Soroye, et al., 2016 [78] | Permanent | 684 | 7.36 | Semi-urban |
| 35. | Sowole, et al., 2007 [79] | Primary | 684 | 10.5 | Semi-urban |

with prevalence ranging from 3.02% to 10.54% [32,46,53,55,65–67,69,70], and there were no studies conducted in the rural setting.

Of the 12 studies in the mixed dentition [35,37,38,44,50,56,57,61,63,67,68,70], 5 were conducted in the urban setting with prevalence ranging from 9.31% to 29.2% [35,44,67,68,70], 2 were conducted in the rural setting with prevalence ranging from 11.1% to 13.0% [38,61], and 5 were conducted in the semi-urban setting with prevalence ranging from 5.20% to 22.9% [37,50,56,57,63].

Of the 19 studies in the permanent dentition [33,37,38,40–43,45,48,58,61,62,64,67,68,70–72,78], 11 were conducted in the urban setting with prevalence ranging from 1.52% to 64.7% [40–43,45,58,62,67,68,70,71], 3 were conducted in the rural setting with prevalence ranging from 1.16% to 35.1% [38,61,64], and 5 were conducted in the semi-urban setting with prevalence ranging from 7.36% to 24.7% [33,37,48,72,78].

## Quantitative analysis

Fig 2 presents the forest plot of the pooled prevalence of dental caries in Nigeria. Based on a random-effects meta-analysis, the overall pooled prevalence was 17% (95% CI: 14%–21%). Substantial heterogeneity was observed among the included studies ($I^2 = 97\%$, $\chi^2 = 1312.83$, $p < 0.01$), indicating considerable variability in prevalence estimates across different study populations, geographic locations, and methodological characteristics. Given the high heterogeneity, the pooled estimate should be interpreted as an average prevalence across diverse study contexts rather than a single uniform national rate.

## Prevalence of dental caries in primary dentition Nigeria

As shown in Fig 3, the pooled prevalence of dental caries in primary dentition was 16% (95% CI: 10%–24%) based on a random-effects model. Substantial heterogeneity was observed among the included studies ($I^2 = 98\%$, $p < 0.01$), indicating considerable variability across study settings and populations. Also, the pooled prevalence of dental caries in mixed dentition was 16% (95% CI: 11%–23%). There was substantial heterogeneity across studies ($I^2 = 97\%$, $p < 0.01$), reflecting differences in population characteristics and study methodology. Also, the pooled prevalence of dental caries in permanent dentition was 20% (95% CI: 16%–26%). High heterogeneity was also observed ($I^2 = 96\%$, $p < 0.01$), suggesting notable between-study variability.

The test for subgroup differences was not statistically significant ($\chi^2 = 1.53$, df = 2, $p = 0.47$), indicating that prevalence did not significantly differ across dentition types.

## Prevalence of dental caries by study setting in Nigeria

As shown in Fig 4, the pooled prevalence of dental caries varied across study settings. In urban settings, the pooled prevalence was 14% (95% CI: 6%–29%), with substantial heterogeneity ($I^2 = 98\%$, $p < 0.01$). In semi-urban settings, the pooled prevalence was 17% (95% CI: 14%–22%), with high heterogeneity ($I^2 = 97\%$, $p < 0.01$). In rural settings, the pooled prevalence was 22% (95% CI: 7%–52%), also with substantial heterogeneity ($I^2 = 98\%$, $p < 0.01$).

Despite apparent differences in point estimates, the test for subgroup differences was not statistically significant ($\chi^2 = 0.43$, df = 2, $p = 0.81$), indicating that prevalence did not significantly differ across study settings.

## Publication bias

Fig 5 present the funnel plots used to assess potential publication bias. Visual inspection suggested some degree of asymmetry. However, Egger's linear regression test did not demonstrate statistically significant evidence of funnel plot asymmetry or small-study effects (t = −0.60, df = 33, $p = 0.552$).

Given the substantial heterogeneity observed across studies ($I^2 \approx 97$–99%), the apparent asymmetry may reflect genuine between-study variability in population characteristics, study settings, and diagnostic criteria rather than true publication bias. Therefore, statistical evidence of publication bias was not detected..

**Fig 2. Prevalence of dental caries in Nigeria.**

## Meta-regression analysis

Fig 6 presents the meta-regression analysis examining publication year as a continuous moderator of dental caries prevalence. The regression model demonstrated a positive but statistically non-significant association between year of publication and prevalence (coefficient=0.0384, p=0.139). Although the regression line indicates a slight upward temporal trend, the wide confidence band and non-significant p-value suggest that publication year did not significantly explain the observed between-study heterogeneity. Therefore, temporal variation was not a significant predictor of pooled prevalence in this analysis.

## Random-effects meta-regression model

Fig 7 displays the random-effects meta-regression forest plot using logit-transformed prevalence estimates. The pooled logit prevalence was −1.74 (95% CI: −1.95 to −1.53), corresponding to a back-transformed prevalence of approximately

**Fig 3. Prevalence of dental caries in primary, mixed and permanent dentition.**

17%. Individual study weights ranged from 1.4% to 1.8%, indicating relatively similar precision across studies. Despite substantial heterogeneity, the pooled estimate remained stable within the confidence interval shown.

## Sensitivity analysis

Fig 8 presents the leave-one-out sensitivity analysis. Sequential exclusion of each individual study did not materially alter the pooled prevalence estimate. The pooled estimate remained stable across iterations, with only minimal fluctuations.

| Study | Events | Total | Prevalence | 95% C.I. |
|---|---|---|---|---|
| `Study Settings` = Urban | | | | |
| Adeniyi 2017 | 87 | 414 | 0.21 | [0.17; 0.25] |
| Dedeke 2014 | 200 | 2149 | 0.09 | [0.08; 0.11] |
| Random effects model | | . | 0.14 | [0.06; 0.29] |
| Heterogeneity: $I^2$ = 98%, $\tau^2$ = 0.4438, $\chi_1^2$ = 45.23 ($p$ < 0.01) | | | | |
| `Study Settings` = Semi-urban | | | | |
| Akinwonmi 2019 | 47 | 206 | 0.23 | [0.18; 0.29] |
| Folayan 2022 | 69 | 1326 | 0.05 | [0.04; 0.07] |
| Kolawole 2019 | 74 | 495 | 0.15 | [0.12; 0.18] |
| Okolo 2022 | 159 | 694 | 0.23 | [0.20; 0.26] |
| Olatosi 2020 | 95 | 592 | 0.16 | [0.13; 0.19] |
| Adekoya-Sofowora 2006 | 56 | 402 | 0.14 | [0.11; 0.18] |
| Ajayi 2015 | 11 | 17 | 0.65 | [0.40; 0.83] |
| Braimoh 2011 | 63 | 140 | 0.45 | [0.37; 0.53] |
| Braimoh 2014 | 30 | 195 | 0.15 | [0.11; 0.21] |
| Chukwumah 2015 | 392 | 1790 | 0.22 | [0.20; 0.24] |
| Denloye 2005 | 65 | 577 | 0.11 | [0.09; 0.14] |
| Ekowmenhenhen 2019 | 83 | 474 | 0.18 | [0.14; 0.21] |
| Lawal 2017 | 55 | 430 | 0.13 | [0.10; 0.16] |
| Okoli 2021 | 275 | 900 | 0.31 | [0.28; 0.34] |
| Olabisi 2015 | 101 | 288 | 0.35 | [0.30; 0.41] |
| Onyejaka 2021 | 45 | 408 | 0.11 | [0.08; 0.14] |
| Osuh 2022 | 335 | 1357 | 0.25 | [0.22; 0.27] |
| Soroye 2016 | 44 | 598 | 0.07 | [0.06; 0.10] |
| Adeniyi 2009 | 44 | 404 | 0.11 | [0.08; 0.14] |
| Denloye 2012 | 7 | 61 | 0.11 | [0.06; 0.22] |
| Folayan 2015 | 33 | 497 | 0.07 | [0.05; 0.09] |
| Folayan 2020 | 51 | 1549 | 0.03 | [0.03; 0.04] |
| Folayan 2020 | 92 | 1155 | 0.08 | [0.07; 0.10] |
| Iyun, 2014 | 127 | 540 | 0.24 | [0.20; 0.27] |
| Kolawole 2016 | 104 | 992 | 0.10 | [0.09; 0.13] |
| Olatosi 2015 | 64 | 302 | 0.21 | [0.17; 0.26] |
| Olatosi 2022 | 122 | 684 | 0.18 | [0.15; 0.21] |
| Olatosi 2022 | 261 | 273 | 0.96 | [0.92; 0.97] |
| Onyejaka 2016 | 42 | 429 | 0.10 | [0.07; 0.13] |
| Onyejaka 2021 | 149 | 657 | 0.23 | [0.20; 0.26] |
| Sowole 2007 | 41 | 389 | 0.11 | [0.08; 0.14] |
| Random effects model | | . | 0.17 | [0.14; 0.22] |
| Heterogeneity: $I^2$ = 97%, $\tau^2$ = 0.5148, $\chi_{30}^2$ = 1146.21 ($p$ < 0.01) | | | | |
| `Study Settings` = Rural | | | | |
| Akinyamoju 2018 | 95 | 778 | 0.12 | [0.10; 0.15] |
| Ogbeide 2022 | 84 | 236 | 0.36 | [0.30; 0.42] |
| Random effects model | | . | 0.22 | [0.07; 0.52] |
| Heterogeneity: $I^2$ = 98%, $\tau^2$ = 0.9363, $\chi_1^2$ = 62.45 ($p$ < 0.01) | | | | |
| **Random effects model** | | . | **0.17** | **[0.14; 0.21]** |
| Heterogeneity: $I^2$ = 97%, $\tau^2$ = 0.5000, $\chi_{34}^2$ = 1312.83 ($p$ < 0.01) | | | | |
| Test for subgroup differences: $\chi_2^2$ = 0.43, df = 2 ($p$ = 0.81) | | | | |

Prevalence of dental caries by study settings(%)

**Fig 4. Prevalence of dental caries in study settings in Nigeria.**

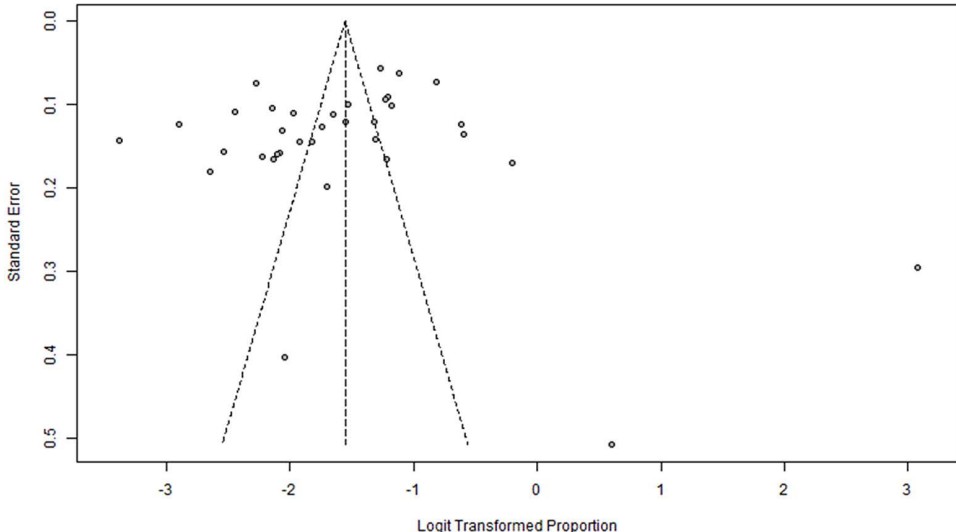

**Fig 5. Funnel plot of comparison.**

Notably, exclusion of the only retrospective/cohort-type study included in the meta-analysis did not materially alter the pooled prevalence estimate, further supporting the robustness of the findings.

Fig 9 illustrates the trim-and-fill analysis conducted to assess the potential impact of small-study effects. Although the method suggested the possible presence of missing studies, the adjusted pooled prevalence (0.22; 95% CI: 0.17–0.26) remained comparable to the original pooled estimate. This suggests that any potential small-study effects or publication bias did not meaningfully affect the overall conclusions of the meta-analysis.

## Discussion

The findings of this study indicate that approximately one in five individuals in Nigeria has dental caries. The highest pooled prevalence was observed in primary dentition, followed by permanent dentition, while the lowest prevalence was recorded in mixed dentition. Dental caries were more common in rural areas than in urban settings and least common in semi-urban areas. Data were available from five of Nigeria's six geopolitical zones, although publications were disproportionately concentrated in Southwestern Nigeria. Overall, studies were conducted in only 10 of the 36 Nigerian states. Over the past 18 years, there has been a steady increase in publications on dental caries prevalence in Nigeria, reflecting growing academic interest and recognition of oral health as a public health concern.

The burden of dental caries in Nigeria remains substantial. However, the pooled estimate should be interpreted in the context of the considerable heterogeneity observed across studies ($I^2 > 95\%$). This variability likely reflects genuine differences in study populations, age distributions, geographic settings, diagnostic criteria, and access to dental services rather than methodological limitations alone. The meta-regression analysis demonstrated that publication year did not significantly explain this heterogeneity, suggesting that contextual and structural factors may be driving the observed variability. Therefore, the pooled estimate represents a summary of diverse epidemiological contexts across Nigeria rather than a single uniform national prevalence.

Similar to other emerging economies, Nigeria's oral healthcare system remains largely curative, with limited implementation of sustained community-based oral health promotion programs [98,99]. Despite this, the pooled prevalence reported in this study is lower than that reported in Ethiopia (40.98%) [100], China (41.15% − 67%) [101], Gulf countries (64.7%) [102], Brazil (72.9%) [103], and Kosovo (72.0%) [104]. Previous reports have also suggested relatively low caries

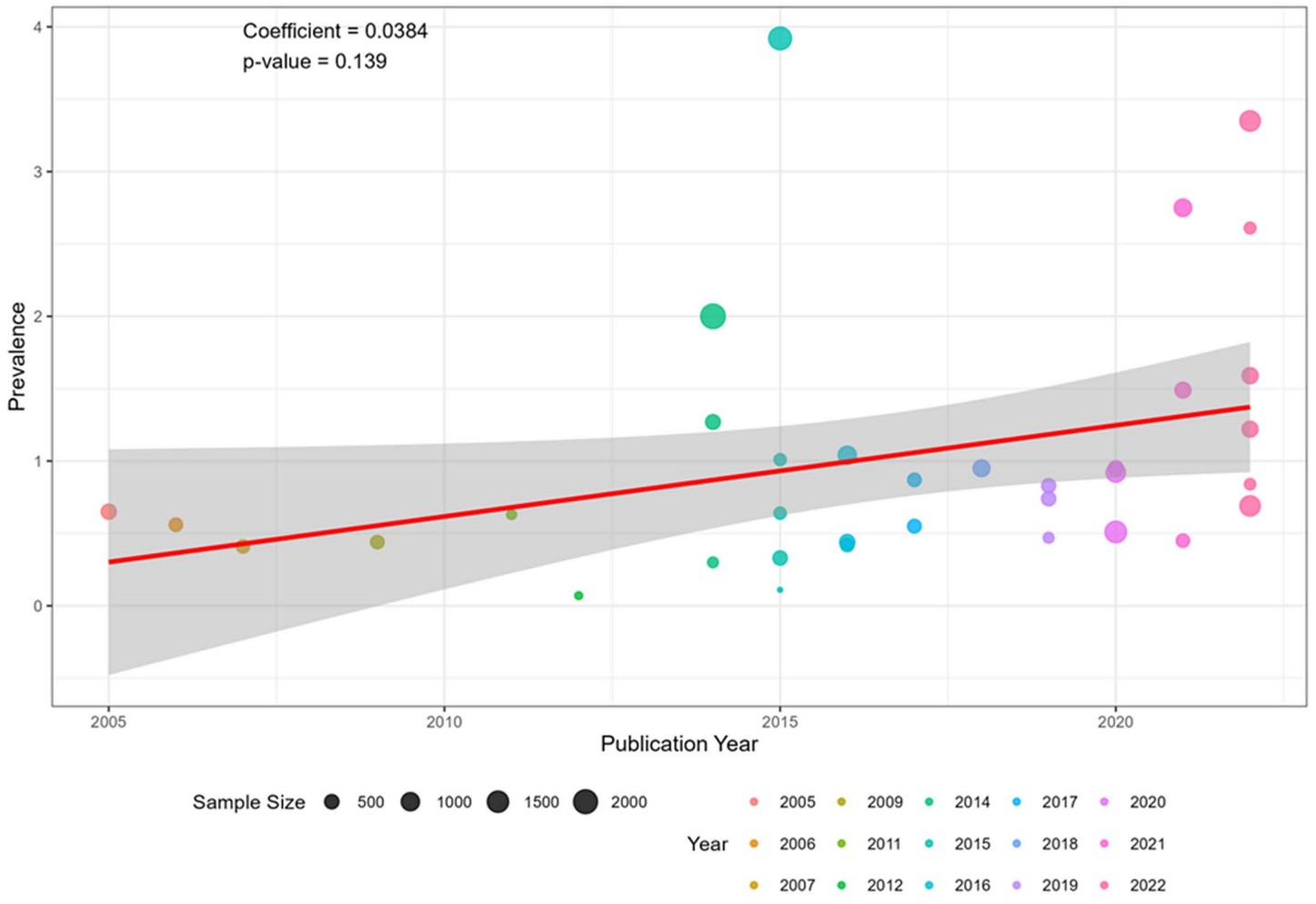

**Fig 6. Meta-regression.**

prevalence in primary dentition in Nigeria [50]. The reasons for this comparatively lower prevalence remain unclear, particularly given Nigeria's similar socioeconomic context to other low- and middle-income countries. One possible explanation may be the reported high use of fluoridated toothpaste [86]. Nonetheless, further epidemiological studies are required to clarify contributory behavioral and environmental factors.

The analysis further showed that caries prevalence varied by dentition type, with the highest prevalence in primary dentition and the lowest in mixed dentition. Although the prevalence in primary dentition was slightly higher than in permanent dentition, the difference was not statistically significant. This contrasts with findings from Eastern Mediterranean countries [105] and East Africa [106], where caries prevalence increases with age. Age-related increases in caries may result from cumulative plaque accumulation, age-related reductions in salivary flow and buffering capacity, gingival recession exposing vulnerable root surfaces, increased lifetime exposure to sugary diets, and degradation of dental restorations over time [107–111]. The similarity between primary and permanent dentition in Nigeria therefore warrants further investigation to identify local determinants that may modify these patterns.

With respect to geographical setting, caries prevalence was higher in rural areas than urban areas and lowest in semi-urban settings. This aligns with evidence highlighting rural challenges, including limited access to dental services,

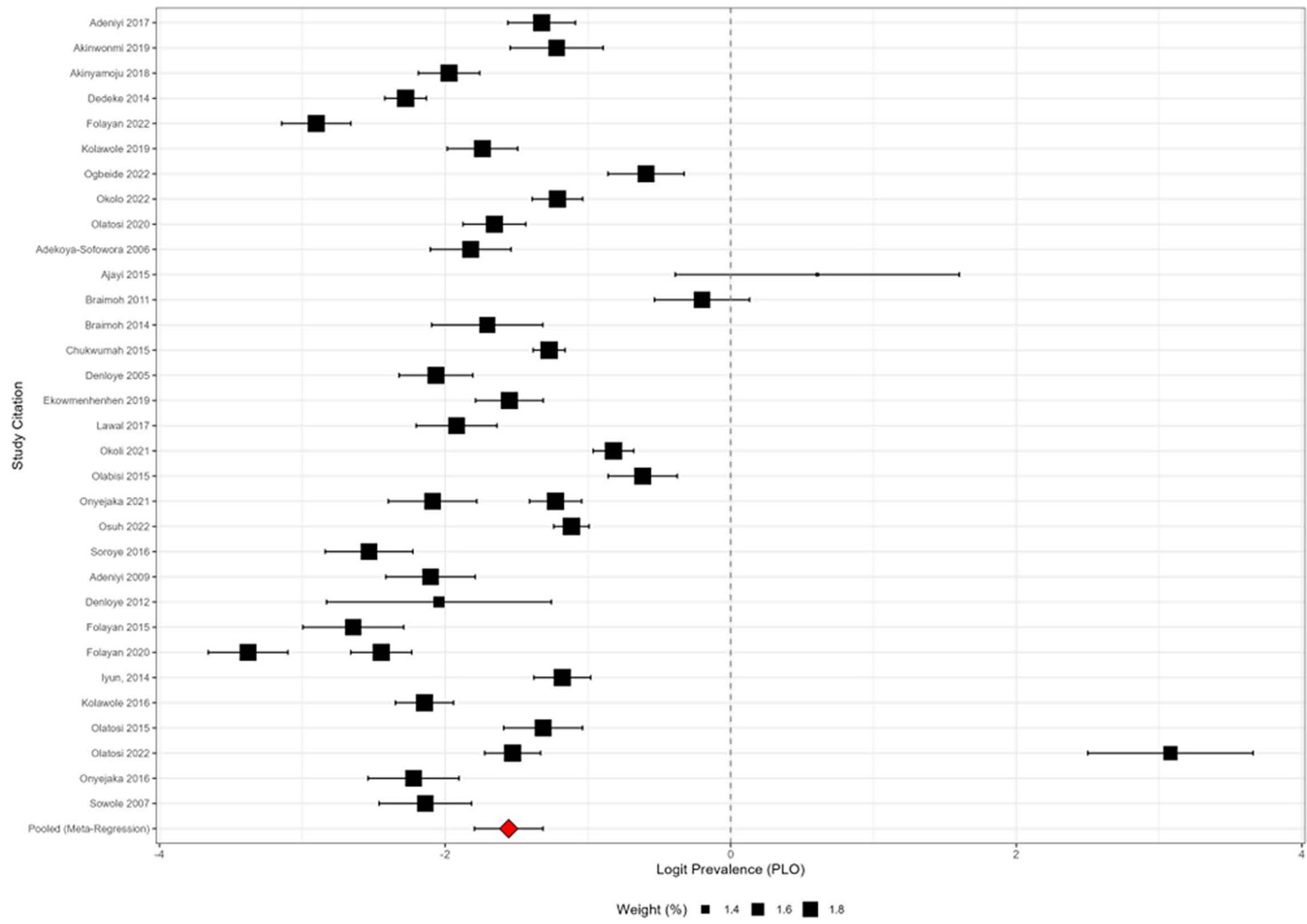

**Fig 7. Random-effects meta-regression model.**

lower oral health literacy, and restricted preventive care [105]. Urban populations may benefit from better access to healthcare facilities and oral health education, contributing to improved outcomes [105]. Similar patterns have been reported in Yemen [112].

Urban–rural disparities may reflect broader social inequalities influencing oral health behaviors and service access. Differences in access to fluoridated water, preventive services, restorative care, and community-level interventions likely contribute to these patterns. Although urbanization has been associated with increased caries risk in some settings, this effect often attenuates after adjusting for socioeconomic status [113]. While the present study indicates lower overall prevalence in urban areas, urban slum populations may remain particularly vulnerable [72]. Future research exploring intra-urban and intra-rural disparities would help identify high-risk subpopulations.

The relatively lower prevalence observed in semi-urban populations may partly reflect contextual factors or reporting bias. Many semi-urban studies were conducted in towns with relatively better access to tertiary dental services and ongoing oral health initiatives. Further research is needed to disentangle the effects of lifestyle, access to care, and socioeconomic conditions across rural, semi-urban, and urban populations in Nigeria.

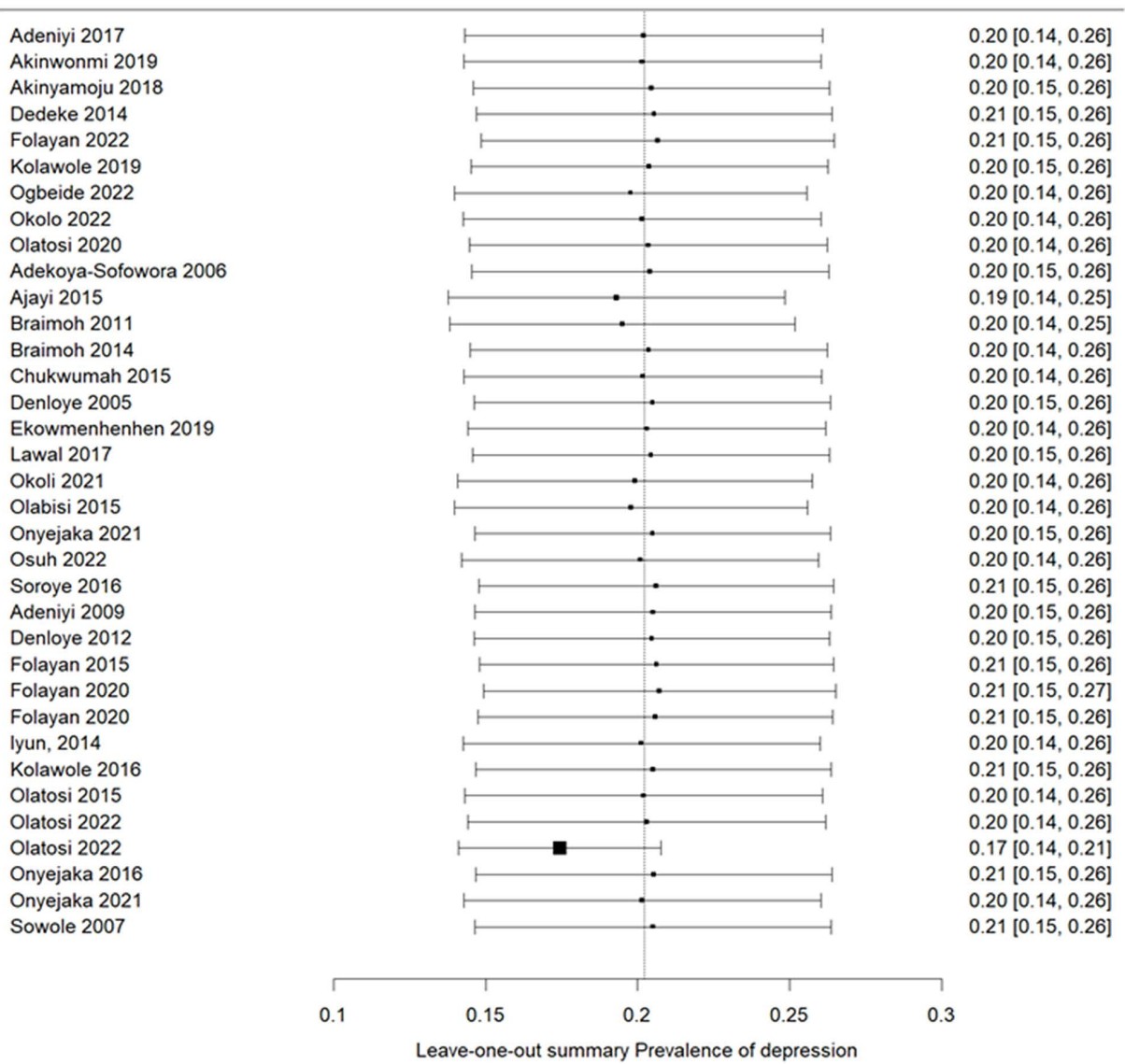

**Fig 8. Leave-one-out sensitivity analysis.**

This study represents the first comprehensive pooled prevalence analysis of dental caries in Nigeria, providing nationally synthesized evidence on disease burden. Strengths include an extensive multi-database search, systematic review and meta-analysis procedures, independent data extraction by four reviewers, absence of language restrictions, and formal quality assessment. Inclusion of low- and moderate-risk studies enhances the credibility and applicability of the findings [114,115].

Nonetheless, limitations should be acknowledged. The overrepresentation of studies from Southwestern Nigeria, where 35.3% of Nigeria's 17 dental schools are located, may limit national generalizability. Although two authors contributed multiple studies, each was conducted in independent populations without overlapping samples, reducing the likelihood of clustering bias at the author level. Although one retrospective hospital-based study was included in the meta-analysis [43], leave-one-out sensitivity analysis demonstrated that its inclusion did not materially influence the pooled estimate;

**Fig 9. Trim-and-fill analysis.**

nevertheless, its clinical context differed from the predominantly cross-sectional studies and should be considered when interpreting the findings. Despite these limitations, this study provides important and updated evidence on the epidemiology of dental caries in Nigeria.

## Conclusion

This systematic review and meta-analysis provides the first nationally synthesized estimate of dental caries prevalence in Nigeria. The pooled prevalence of approximately 17% indicates that dental caries remains a significant public health concern

in the country. Although variations were observed across dentition types and geographical settings, these differences were not statistically significant, and substantial heterogeneity reflects the diverse epidemiological contexts across Nigeria. These findings underscore the need for strengthened preventive strategies, improved access to oral healthcare services, and context-specific public health interventions to address the burden of dental caries nationwide. Further high-quality epidemiological studies are required to better understand regional disparities and determinants of caries risk in Nigeria.

## Supporting information

**S1 File. PRISMA Checklist.**
(PDF)

**S2 File. Search Strategy.**
(PDF)

**S3 File. Excluded Studies with their Reasons.**
(XLSX)

**S1 Table. Quality and Risk of Bias Assessment.**
(PDF)

**S2 Table. GRADE Rating Quality of Evidence.**
(PDF)

## Author contributions

**Conceptualization:** Folahanmi Tomiwa Akinsolu, George Uchenna Eleje, Joanne Lusher, Foluso Owotade, Oliver Chukwujekwu Ezechi, Morẹ́nikẹ́ Oluwátóyìn Foláyan.

**Data curation:** Folahanmi Tomiwa Akinsolu, Olunike Rebecca Abodunrin, Abel Chukwuemeka, Mobolaji Timothy Olagunju, Ifeoluwa Eunice Adewole, George Uchenna Eleje, Adebola Oluyemisi Ehizele, Morẹ́nikẹ́ Oluwátóyìn Foláyan.

**Formal analysis:** Folahanmi Tomiwa Akinsolu, Olunike Rebecca Abodunrin, Abel Chukwuemeka, Mobolaji Timothy Olagunju, Ifeoluwa Eunice Adewole, George Uchenna Eleje, Adebola Oluyemisi Ehizele.

**Funding acquisition:** Oliver Chukwujekwu Ezechi.

**Investigation:** Mobolaji Timothy Olagunju, Morẹ́nikẹ́ Oluwátóyìn Foláyan.

**Methodology:** Folahanmi Tomiwa Akinsolu, Olunike Rebecca Abodunrin, Abel Chukwuemeka, Ifeoluwa Eunice Adewole, George Uchenna Eleje, Oliver Chukwujekwu Ezechi, Morẹ́nikẹ́ Oluwátóyìn Foláyan.

**Project administration:** Folahanmi Tomiwa Akinsolu, Mobolaji Timothy Olagunju.

**Software:** Mobolaji Timothy Olagunju.

**Supervision:** Oliver Chukwujekwu Ezechi, Morẹ́nikẹ́ Oluwátóyìn Foláyan.

**Validation:** Folahanmi Tomiwa Akinsolu, Olunike Rebecca Abodunrin, Oliver Chukwujekwu Ezechi, Morẹ́nikẹ́ Oluwátóyìn Foláyan.

**Visualization:** Folahanmi Tomiwa Akinsolu, Mobolaji Timothy Olagunju, Oliver Chukwujekwu Ezechi, Morẹ́nikẹ́ Oluwátóyìn Foláyan.

**Writing – original draft:** Folahanmi Tomiwa Akinsolu, Olunike Rebecca Abodunrin, Mobolaji Timothy Olagunju, Ifeoluwa Eunice Adewole, George Uchenna Eleje, Adebola Oluyemisi Ehizele, Joanne Lusher, Foluso Owotade, Oliver Chukwujekwu Ezechi, Morẹ́nikẹ́ Oluwátóyìn Foláyan.

**Writing – review & editing:** Folahanmi Tomiwa Akinsolu, Olunike Rebecca Abodunrin, Mobolaji Timothy Olagunju, Abideen Olurotimi Salako, George Uchenna Eleje, Adebola Oluyemisi Ehizele, Joanne Lusher, Foluso Owotade, Oliver Chukwujekwu Ezechi, Mórénikẹ́ Oluwátóyìn Foláyan.

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
