## [Decision Letter · Decision Letter 0]

8 Aug 2024

PONE-D-24-23202Prevalence of Dental Caries in the Primary, Mixed and Permanent Dentitions in Nigeria: A Systematic Review and Meta-analysisPLOS ONE

Dear Dr. Akinsolu,

Thank you for submitting your manuscript to PLOS ONE. After careful consideration, we feel that it has merit but does not fully meet PLOS ONE’s publication criteria as it currently stands. Therefore, we invite you to submit a revised version of the manuscript that addresses the points raised during the review process.

We look forward to receiving your revised manuscript.

Kind regards,

Yolanda Malele-Kolisa, BDS, MPH, MDent, PhD

Academic Editor

PLOS ONE

Journal Requirements:

2. Thank you for stating the following financial disclosure: "Grant Number: 5NM-ADJGT-22-0082 from the Nigerian Institute for Medical Research"

3. Thank you for stating the following in the Acknowledgments Section of your manuscript: "The authors thank the Nigerian Institute for Medical Research for supporting the study. "

Please remove any funding-related text from the manuscript and let us know how you would like to update your Funding Statement. Currently, your Funding Statement reads as follows: "Grant Number: 5NM-ADJGT-22-0082 from the Nigerian Institute for Medical Research"

Additional Editor Comments :

The Systematic Review requires major revision. Systematic reviews provide the best available evidence to change and influence policy and practice. The following methodology changes are proposed:

1. The search words used for review should be specified.

2. Specify Data Synthesis process

3. Research question highlighting the phenomenon of interest, population, outcome measure in addition to the stated language, study types etc.

4. For statistical pooling, specify the justification for using the random effect model.

5. Address the Reviewer comments

Reviewers' comments:

Reviewer's Responses to Questions

**Comments to the Author**

1. Is the manuscript technically sound, and do the data support the conclusions?

Reviewer #1: Partly

2. Has the statistical analysis been performed appropriately and rigorously? 

Reviewer #1: No

3. Have the authors made all data underlying the findings in their manuscript fully available?

Reviewer #1: Yes

4. Is the manuscript presented in an intelligible fashion and written in standard English?

Reviewer #1: Yes

5. Review Comments to the Author

Reviewer #1: CORRECTION OF THE MANUSCRIPT (PONE-D-24-23202)

PLOS ONE

I am submitting my comments regarding the manuscript titled "Prevalence of Dental Caries in the Primary, Mixed and Permanent Dentitions in Nigeria: A Systematic Review and Meta-analysis." The study is a systematic review of prevalence aimed at identifying the prevalence of dental caries in Nigeria across deciduous, mixed, and permanent dentitions. I believe this study is significant. However, I list some concerns that should be considered by the authors.

In the introduction, there is a need for a better justification of the study's publication. What is its importance to the scientific community? How does it benefit the population of Nigeria? Have other countries conducted similar analyses?

In the methodology, please include the research question and the acronym used to outline the study. To assess bias risk, I suggest using the Joanna Briggs Institute's tool. Additionally, I recommend using the GRADE checklist to assess the certainty of evidence.

In the results, the authors identify extreme prevalence percentages. These data should be interpreted cautiously considering the certainty of evidence from the included studies.

In the discussion, I suggest including the strengths and limitations of the study towards the end of this section. Additionally, all discussed points need to be referenced.

Decision: major review

6. PLOS authors have the option to publish the peer review history of their article (what does this mean?). If published, this will include your full peer review and any attached files.

Reviewer #1: No

---

## [Author Response · Author response to Decision Letter 1]

24 Nov 2024

Reviewer

Dear Reviewer,

We would like to express our sincere gratitude for the thorough review of our manuscript (PONE-D-24-23202) titled “Prevalence of Dental Caries in the Primary, Mixed, and Permanent Dentitions in Nigeria: A Systematic Review and Meta-analysis.”

We have carefully addressed each comment and made necessary revisions to resolve the concerns raised.

We are thankful for your constructive input.

Below are the responses to each of the comments:

1. In the introduction, there is a need for a better justification of the study's publication. What is its importance to the scientific community? How does it benefit the population of Nigeria? Have other countries conducted similar analyses?

Response

This concern is addressed in detail in lines 98 to 107,

“This study addresses a significant gap in understanding the prevalence and severity of dental caries in Nigeria. Given the increasing burden of dental caries in LMICs, particularly in Nigeria, where access to dental care is limited, this research is critical for informing public health interventions. By synthesizing data from various regions and comparing prevalence across urban, rural, and semi-urban settings, the study provides valuable insights into the geographic and demographic disparities in dental caries, offering evidence that can guide policy decisions and resource allocation to improve oral health outcomes in Nigeria. This systematic review and meta-analysis contribute substantially to the global understanding of dental caries, especially in under-researched populations. This study adds to the existing literature by offering Nigeria-specific data critical for developing context-appropriate oral health policies.”

2. In the methodology, please include the research question and the acronym used to outline the study. To assess bias risk, I suggest using the Joanna Briggs Institute's tool. Additionally, I recommend using the GRADE checklist to assess the certainty of evidence.

Response

The research question has been clearly stated in the methodology on lines 119 to 122:

Research Question

What is the prevalence and severity of dental caries in the primary, mixed, and permanent dentition among residents in Nigeria, and how does this vary by geographical location (urban, rural, semi-urban)?

The acronym used to outline the study has been included in lines 123-127

The PICO framework was used to understand the prevalence and severity of dental caries in Nigeria. The Table below shows the components of the PICO framework used for this study.

Table 1: PICO Component

Component Description

Population (P) Residents of Nigeria (children and adults) in urban, rural, and semi-urban areas

Intervention (I) Not applicable (Observational study)

Comparison (C) Comparison between different dentition types (primary, mixed, permanent) and geographic settings (urban, rural, semi-urban)

Outcome (O) Prevalence and severity of dental caries

The risk of bias assessed using the Joanna Briggs Institute Tool and the GRADE checklist to assess the certainty of evidence have been conducted and the details can be S3 and S4 Files respectively. (Lines 167-197)

3. In the results, the authors identify extreme prevalence percentages. These data should be interpreted cautiously considering the certainty of evidence from the included studies.

Response

Based on the GRADE checklist outcome of the included articles, the authors had reasonable certainty that the reported prevalence had moderate certainty of evidence.

4. In the discussion, I suggest including the strengths and limitations of the study towards the end of this section. Additionally, all discussed points need to be referenced.

The strengths and limitations of the study are thoroughly discussed in lines 473 to 489 of the discussion section. Additionally, all points in this section have been properly referenced to ensure a comprehensive and credible analysis.

“This study is the first pooled prevalence analysis of dental caries in Nigeria, providing valuable insights into the condition's prevalence nationwide. A notable strength of the study is the comprehensive search across multiple databases for relevant studies, which were then systematically reviewed and meta-analyzed. To minimize errors, four reviewers performed data extraction independently. Additionally, no language restrictions were applied during the study selection process. The quality assessment, which evaluated the risk of bias within the included studies, demonstrates the overall robustness of the synthesized evidence. The quality assessment results underpin the systematic review's credibility, reflecting a careful selection of studies with varying risk levels. Both low- and moderate-risk studies offer a comprehensive understanding of dental caries prevalence in Nigeria, providing evidence that can be confidently translated into practice [98, 99].

However, the study does have a few limitations. The skewness of data to Southwest Nigeria, where 35.3% of the 17 dental schools in Nigeria are located, could potentially introduce bias in the generalizability of the study. In addition, two authors authored 10 (19.2%) of the studies included in this review, which might further introduce author bias. Despite these possible limitations, the present study does provide substantial and new evidence on the epidemiology of dental caries in Nigeria.”

---

## [Decision Letter · Decision Letter 1]

14 Sep 2025

PONE-D-24-23202R1Prevalence of Dental Caries in the Primary, Mixed and Permanent Dentitions in Nigeria: A Systematic Review and Meta-analysisPLOS ONE

Dear Dr. Akinsolu,

Thank you for submitting your manuscript to PLOS ONE. After careful consideration, we feel that it has merit but does not fully meet PLOS ONE’s publication criteria as it currently stands. Therefore, we invite you to submit a revised version of the manuscript that addresses the points raised during the review process.

We look forward to receiving your revised manuscript.

Kind regards,

Ashish Wasudeo Khobragade, MD

Academic Editor

PLOS ONE

Journal Requirements:

Additional Editor Comments:

1. The % symbol is repeated in each row in Table 3 under the column ‘Prevalence of dental caries’.

2. Please provide high-resolution figures. Most of the figures are blurred and not clearly visible.

3. Comment on the publication bias considering the funnel plot. Provide the results of Egger’s test or Begg and Mazumdar’s test.

4. Studies included in the metanalysis from 2001 to 2023. The prevalence of dental caries varies over the years. How may this affect the pooled prevalence? Provide meta-regression findings.

5. How the sensitivity analysis was conducted, and mention the findings.

Reviewers' comments:

Reviewer's Responses to Questions

**Comments to the Author**

1. If the authors have adequately addressed your comments raised in a previous round of review and you feel that this manuscript is now acceptable for publication, you may indicate that here to bypass the “Comments to the Author” section, enter your conflict of interest statement in the “Confidential to Editor” section, and submit your "Accept" recommendation.

Reviewer #2: (No Response)

Reviewer #3: (No Response)

Reviewer #4: (No Response)

2. Is the manuscript technically sound, and do the data support the conclusions?

Reviewer #2: (No Response)

Reviewer #3: Partly

Reviewer #4: Yes

3. Has the statistical analysis been performed appropriately and rigorously? 

Reviewer #2: (No Response)

Reviewer #3: Yes

Reviewer #4: Yes

4. Have the authors made all data underlying the findings in their manuscript fully available?

Reviewer #2: (No Response)

Reviewer #3: Yes

Reviewer #4: Yes

5. Is the manuscript presented in an intelligible fashion and written in standard English?

Reviewer #2: (No Response)

Reviewer #3: No

Reviewer #4: Yes

6. Review Comments to the Author

Reviewer #2: (No Response)

Reviewer #3: The manuscript presents a systematic review and meta-analysis aimed at determining the prevalence of dental caries in primary, mixed, and permanent dentitions among residents in urban, rural, and semi-urban Nigeria. I think this manuscript is poorly written in many ways, including grammar, English usage, etc., and it cannot be published in this format. The comments are as follows:

1.There are many long sentences that are not easy to understand. The English is weak and should be edited. Some parts are not completely clear. Even the fonts and formatting are not consistent (e.g., L69).

2.The search dates are not consistent. In the abstract (L51), the dates are between 2005 and 2022. In the “Search Strategy and Selection of Studies” section, the dates are between January 2001 and December 2023 (L130).

3.It is necessary to rewrite the discussion by mentioning the important results of the work, quantitatively.

4.Please double-check the reference list and make the formatting consistent.

Reviewer #4: Thank you for the opportunity to review this manuscript. I have some comments which I hope will help improve it.

1. In my opinion, it would be interesting to study the possible evolution of caries prevalence over time. To do this, I suggest two approaches.

1.1. One way to assess the evolution of the effect size (in this case, prevalence) is ordering the studies in the forest plot (Figure 2, at least) in a chronological manner, to visually assess the evolution.

1.2. An alternative would be, of course, performing a meta-regression with the study year as the predictor and see what happens.

2. Figure 9 contains a funnel plot to assess the publication bias. The manuscript says that the studies included are generally precise, but the purpose of the funnel plot is not only that, but also assess publication bias. I think some bias is visible, because for less precise studies only higher prevalences are shown. The funnel plot should be symetrical to show an absence of publication bias, and I do not think that is the case. Anyway, an Egger's test should clarify the question.

3. In the discussion it is said that there are differences between dentition modes, but the confidence intervals obtained overlap in all cases. Same thing happens with the differences between rural and urban areas.

4. The manuscrips points out as a limitation that two authors authored roughly 20% of the studies, which may be a source of bias. I agree. To account for that, I suggest performing a three-level meta-analysis model, adding a level with clusters of studies performed by the same authors. Perhaps this tutorial (10.20982/tqmp.12.3.p154) may be helpful.

7. PLOS authors have the option to publish the peer review history of their article (what does this mean?). If published, this will include your full peer review and any attached files.

Reviewer #2: No

Reviewer #3: No

Reviewer #4: No

---

## [Author Response · Author response to Decision Letter 2]

28 Feb 2026

Response to Reviewers

We sincerely thank the Academic Editor and the Reviewers for their thorough evaluation of our manuscript and for the constructive comments provided. We have carefully revised the manuscript in response to all comments raised. Below, we provide a detailed point-by-point response.

Additional Editor Comments

1. The % symbol is repeated in each row in Table 3 under the column ‘Prevalence of dental caries’.

Response: Thank you for this observation. We have revised Table 3 by removing the repeated “%” symbol in the 'Prevalence of dental caries' column to improve clarity and formatting consistency.

2. Please provide high-resolution figures. Most of the figures are blurred and not clearly visible.

Response: Thank you for this important comment. All figures have now been regenerated and uploaded in high resolution to ensure clarity and readability. In addition, following the reviewers’ suggestions, we reanalyzed the data using R (RStudio) for the meta-regression, Egger’s test, and sensitivity analyses. The updated, high-quality figures reflecting these analyses have been included in the revised manuscript.

3. Comment on the publication bias considering the funnel plot. Provide the results of Egger’s test or Begg and Mazumdar’s test.

Response: Thank you for this important suggestion. We performed Egger’s linear regression test to formally assess funnel plot asymmetry. The test did not indicate statistically significant evidence of small-study effects (t = –0.60, df = 33, p = 0.552). Although visual inspection of the funnel plot suggested some asymmetry, the substantial heterogeneity observed across studies (I² ≈ 97–99%) likely explains the dispersion rather than publication bias. The manuscript has been updated accordingly.

4. Studies included in the meta-analysis from 2001 to 2023. The prevalence of dental caries varies over the years. How may this affect the pooled prevalence? Provide meta-regression findings.

Response: Thank you for this important observation. To assess whether temporal variation influenced the pooled prevalence, we conducted a meta-regression analysis using publication year as a continuous moderator. The analysis demonstrated a positive but non-statistically significant association (coefficient = 0.0384, p = 0.139), indicating that publication year did not significantly explain heterogeneity in prevalence estimates. Therefore, pooling studies across the study period is unlikely to have substantially biased the overall prevalence estimate. The manuscript has been revised to include these findings.

5. How was the sensitivity analysis conducted and what were the findings?

Response: Thank you for this important suggestion. Sensitivity analyses were conducted using two approaches. First, a leave-one-out analysis was performed, whereby each study was sequentially excluded to assess its influence on the pooled estimate. The pooled prevalence remained stable across iterations, indicating that no single study unduly influenced the results. Second, a trim-and-fill procedure was applied to assess the impact of potential small-study effects. The adjusted pooled prevalence remained comparable to the original estimate (0.22; 95% CI: 0.17–0.26), suggesting that the overall findings are robust.

Reviewer #3 Comments

1. The English is weak, sentences are long, and formatting is inconsistent.

Response: We appreciate this observation. The entire manuscript has been thoroughly revised to improve grammar, clarity, and readability. Long and complex sentences have been rewritten, and formatting inconsistencies, including font style and size discrepancies, have been corrected to ensure uniform presentation throughout the manuscript.

2. The search dates are not consistent between the abstract and methods section.

Response: Thank you for identifying this inconsistency. The discrepancy resulted from earlier draft versions. The search dates have now been standardized throughout the manuscript. The final search period is consistently reported as January 2001 to December 2023 in both the abstract and the methods section.

3. The discussion should mention important quantitative results.

Response: Thank you for this constructive suggestion. The Discussion section has been substantially revised to explicitly highlight the key quantitative findings, including the overall pooled prevalence (17%; 95% CI: 14%–21%), subgroup estimates by dentition type and geographical setting, and the extent of heterogeneity (I² values).

4. Please double-check the reference list formatting.

Response: The reference list has been carefully reviewed and reformatted to ensure full consistency with the journal’s citation style. In-text citations and references have been cross-checked for completeness and accuracy.

Reviewer #4 Comments

1. Assess the evolution of prevalence over time using chronological ordering or meta-regression.

Response: We thank the reviewer for this valuable suggestion. To formally assess temporal trends, we conducted a meta-regression analysis using publication year as a continuous moderator variable. The analysis demonstrated a positive but non-statistically significant association (coefficient = 0.0384, p = 0.139), indicating that publication year did not significantly explain between-study heterogeneity.

2. Funnel plot asymmetry suggests possible publication bias. Egger’s test should clarify.

Response: We agree that visual inspection of the funnel plot suggested some asymmetry. To formally assess small-study effects, we conducted Egger’s linear regression test, which did not demonstrate statistically significant evidence of funnel plot asymmetry (t = –0.60, df = 33, p = 0.552). Given the substantial heterogeneity (I² ≈ 97–99%), the asymmetry likely reflects genuine variability rather than publication bias. The manuscript has been revised accordingly.

3. Differences between dentition modes and settings have overlapping confidence intervals.

Response: We agree that the confidence intervals overlap across dentition types and settings. A formal test for subgroup differences was conducted and did not demonstrate statistically significant differences (χ² = 1.53, df = 2, p = 0.47 for dentition type). The Discussion has been revised to clarify that these represent descriptive patterns rather than statistically confirmed differences.

4. Consider a three-level meta-analysis due to clustering by authors.

Response: We thank the reviewer for this thoughtful suggestion. Although two authors contributed multiple studies, each study was conducted in independent populations without overlapping samples. Therefore, the effect sizes represent independent observations. Given the absence of dependency at the dataset level, we believe a three-level meta-analysis model is not required. This clarification has been added to the limitations section.

We hope that the revisions made have adequately addressed all concerns raised by the Editor and Reviewers. We sincerely appreciate the time and expertise invested in reviewing our work.

---

## [Decision Letter · Decision Letter 2]

12 Apr 2026

PONE-D-24-23202R2Prevalence of Dental Caries in the Primary, Mixed and Permanent Dentitions in Nigeria: A Systematic Review and Meta-analysisPLOS One

Dear Dr. Akinsolu,

Thank you for submitting your manuscript to PLOS ONE. After careful consideration, we feel that it has merit but does not fully meet PLOS ONE’s publication criteria as it currently stands. Therefore, we invite you to submit a revised version of the manuscript that addresses the points raised during the review process.

We look forward to receiving your revised manuscript.

Kind regards,

Ashish Wasudeo Khobragade, MD

Academic Editor

PLOS One

Journal Requirements:

Additional Editor Comments:

1. Cohort studies are included in the analysis. The objective is to determine the prevalence of dental caries in the primary, mixed, and permanent dentitions.

2. Recheck the extracted values of prevalence and sample size of all studies included in the meta-analysis.

Reviewers' comments:

Reviewer's Responses to Questions

**Comments to the Author**

1. If the authors have adequately addressed your comments raised in a previous round of review and you feel that this manuscript is now acceptable for publication, you may indicate that here to bypass the “Comments to the Author” section, enter your conflict of interest statement in the “Confidential to Editor” section, and submit your "Accept" recommendation.

Reviewer #2: (No Response)

Reviewer #4: All comments have been addressed

2. Is the manuscript technically sound, and do the data support the conclusions?

Reviewer #2: (No Response)

Reviewer #4: Yes

3. Has the statistical analysis been performed appropriately and rigorously? 

Reviewer #2: (No Response)

Reviewer #4: Yes

4. Have the authors made all data underlying the findings in their manuscript fully available?

Reviewer #2: (No Response)

Reviewer #4: Yes

5. Is the manuscript presented in an intelligible fashion and written in standard English?

Reviewer #2: (No Response)

Reviewer #4: Yes

6. Review Comments to the Author

Reviewer #2: (No Response)

Reviewer #4: (No Response)

7. PLOS authors have the option to publish the peer review history of their article (what does this mean?). If published, this will include your full peer review and any attached files.

Reviewer #2: No

Reviewer #4: No

---

## [Author Response · Author response to Decision Letter 3]

19 Apr 2026

Response to Academic Editor

Comment 1

Cohort studies are included in the analysis. The objective is to determine the prevalence of dental caries in the primary, mixed, and permanent dentitions.

Response:

Thank you for this important observation. We carefully re-examined the study designs of all studies included in the meta-analysis and confirmed that only one retrospective/cohort-type study was included. This study reported extractable prevalence-related data at a defined assessment point relevant to the review objective and was therefore retained in the analysis.

To ensure that its inclusion did not bias the pooled estimate, we conducted a leave-one-out sensitivity analysis. The results showed that exclusion of this study did not materially alter the overall pooled prevalence, indicating that it did not exert a disproportionate influence on the findings.

We have clarified this in the Methods, Results, and Limitations sections of the revised manuscript to improve transparency.

Comment 2

Recheck the extracted values of prevalence and sample size of all studies included in the meta-analysis.

Response:

Thank you for this important comment. We conducted a comprehensive re-evaluation of all extracted data, including prevalence values, sample sizes, dentition categories, and study setting classifications for each study included in the meta-analysis.

---

## [Editor Report · Decision Letter 3]

27 Apr 2026

Prevalence of Dental Caries in the Primary, Mixed and Permanent Dentitions in Nigeria: A Systematic Review and Meta-analysis

PONE-D-24-23202R3

Dear Dr. Akinsolu,

We’re pleased to inform you that your manuscript has been judged scientifically suitable for publication and will be formally accepted for publication once it meets all outstanding technical requirements.

Kind regards,

Ashish Wasudeo Khobragade, MD

Academic Editor

PLOS One
---

## [Editor Report · Acceptance letter]

PONE-D-24-23202R3

PLOS One

Dear Dr. Akinsolu,

I'm pleased to inform you that your manuscript has been deemed suitable for publication in PLOS One. Congratulations! Your manuscript is now being handed over to our production team.

Kind regards,

on behalf of

Dr. Ashish Wasudeo Khobragade

Academic Editor

PLOS One